# Changes in Traditional Activities of Industrial Area toward Sustainable Tourism Development

**Milena Nedeljković Knežević [1], Marko D. Petrović [2,3],[\*] , Slađana Nedeljković [4], Maja Mijatov [1], Milan M. Radovanović [2,3], Mirjana Gajić [5] and Miroslav Škoda [6]**

[1]   Faculty of Sciences, Department of Geography, Tourism and Hotel Management, University of Novi Sad, 21000 Novi Sad, Serbia; milena.nedeljkovic@dgt.uns.ac.rs (M.N.K.); majam@dgt.uns.ac.rs (M.M.)
[2]   Geographical Institute "Jovan Cvijić" SASA, 11000 Belgrade, Serbia; m.radovanovic@gi.sanu.ac.rs
[3]   Institute of Sports, Tourism and Service, South Ural State University, 454080 Chelyabinsk, Russia
[4]   EPS, RB Kolubara, Svetog Save 1, 11550 Lazarevac, Serbia; sladjana.nedeljkovic@eps.rs
[5]   Faculty of Geography, University of Belgrade, 11000 Belgrade, Serbia; mgajic@gef.bg.ac.rs
[6]   Department of Economics and Management, DTI University, 01841 Dubnicanad Váhom, Slovakia; Skoda@dti.sk
[\*]   Correspondence: m.petrovic@gi.sanu.ac.rs; Tel.: +381-6417-482-57

**Abstract:** The purpose of this study is to investigate the potential for restructuring industrial areas toward tourism development within local communities, with a special emphasis on the socio-cultural determinants of residents, as well as their attitudes regarding the sustainable development of tourism. The research is also oriented toward the interests of local communities with respect to entrepreneurial activities in the field of tourism within regions relying on traditional industries, in this case, one of the largest open-pit mining surfaces in Europe (near the Serbian town of Lazarevac). The survey was conducted on a sample of 273 respondents. The research results point to the residents' attitudes regarding the acceptability of tourism development options, as well as their attitudes toward tourism development, with the aim of providing the conditions for a successful transition from a typical heavy industrial setting toward sustainable tourism development.

**Keywords:** sustainable development; industrial environment; local community; tourism; locals' attitudes

## 1. Introduction

One of the main characteristics of traditional industrial regions is their high degree of agglomeration, or, more precisely, the intensive concentration of production structures, which, in the long period of development of these regions, was considered to be one of the most important advantages in achieving business success [1]. However, over time, in numerous similar regions, organizational structures were highly resistant to the necessity of carrying out organizational change, especially in transitional countries, where these regions were under direct state control, providing a highly monopolistic position for these organizations in the market [2]. The accumulation of the abovementioned specificities in traditional industrial regions and their extensive reliance on one type of industry (in this case, on energy resources) became an obstacle to competitiveness in these regions, especially given the fact that there are problems related to a surplus of employees, who were not able to receive "training" in internal entrepreneurship in the organizations in which they were employed [3]. According to previous findings, traditional industrial regions could be classified as "second-league" regions, based on the performances achieved by organizations operating in these regions, but also with respect to the fact that they are not so familiar with developmental trends [4,5]. As a further

consequence of this, their involvement in the global economic competitiveness of the region will be handicapped by their inconvenient starting positions.

A good practice for successful transformation processes involves the usage of local industrial traditions in order to build a new image of the region to be transformed [6]. Due to the fact that tourism represents the third biggest "industry" in the world (right after the oil and car industries), according to the amount of generated revenue, numerous countries that possess natural and cultural resources suitable for inclusion in tourism development treat this activity as a strategic direction for further economic development [7]. Therefore, numerous researchers regarded successful tourism development as being mainly dependent on the cooperation of different segments of the local community throughout the sustainable planning process [8–13]. The fact that members of the local community are key stakeholders in commencing entrepreneurial activities in tourism indicates the importance of researching their attitudes regarding the possibilities of tourism development [14]. Residents' attitudes are particularly important, considering the fact that members of the local community could be less supportive of tourism if it decreases their quality of life [15]. This approach to the process of planning tourism development is also based on the well-known stakeholder approach in strategic management (which indicates the necessity of balancing the interests of various stakeholders in the local community), as well as on sociological theories of social exchange [16–25]. According to the research conducted by Perić, Đurkin, and Wise [26], in cases where local community members represent the majority of the event's active and passive participants, the fostering of business networking may be able to finally enhance the involvement of local businesses in playing an important role in ensuring that there is a sense of cohesion among community and stakeholders.

For the purposes of this research, the concept of a local community is defined on the basis of its territory (encompassing Lazarevac municipality), in sociological terms (it consists of the people living within this territory, in accordance with cultural, historical, and traditional values), as well as in institutional terms (on the basis of the structure of the local self-government). This research was conducted in the local community of Lazarevac municipality (the wider territory of Serbian capital city of Belgrade), or, more precisely, at the site of Kolubara mine, one of the largest mine surfaces in Europe, which employs about one-third of the local residents. Plans for the development of this community predict extensive restructuring, including privatization and significant reductions in the number of employees. Development planners of this municipality must consider alternative development directions, especially those belonging to the service sector, including tourism. This means of development proved to be effective in a number of different foreign industrial regions, due to the fact that it caused an increase in the rate of employment and in the diversification of economic activities. One of the key positive outcomes of tourism development in the local community is the influence of this sector in terms of increasing the economic benefits for residents [16]. In this study, the research objectives were as follows:

- Identification of the socio-cultural determinants of attitudes among the local community regarding economic and non-economic profit and the costs of tourism development in traditional industrial regions of Serbia;
- Evaluation of the readiness of the local residents to participate in entrepreneurial activities in the tourism sector.

Research into these objectives is important, due to the fact that they could provide specific proposals for the absorption of the labor "surplus", which occurs as a result of the entire restructuring process of this region. These changes, related to reorientation from traditional industry toward sustainable tourism development, occur as a consequence of technological progress, on one hand, together with the global trends that strengthen the service sector, on the other.

## 2. Literature Review

Studies on the residents' attitudes regarding directions of development of the local community are treated as interdisciplinary studies in the field of environmental sociology (natural and social), and this field of sociology was especially developed over the last three decades [27–32]. Research on the residents' attitudes toward tourism development, including their willingness to participate in entrepreneurial activities in the tourism sector, was previously conducted and presented in numerous studies [33–37]. The attitudes of the local community are typically strongly associated with deeply rooted values, as well as with personalities which are not so prone to changes [38–42]. The role of the members of the local community might be important throughout their voluntary engagement, oriented toward further development of the local community. In the research conducted by Wise, Perić, and Đurkin [43], the authors emphasized the importance of the members of the local community in the realization of events, based on the fact that their volunteer activities might contribute to visitors' feeling of safety and agreeableness. According to the same study, the local community could encourage visitors to stay in the local area. If local communities are given the opportunity to actively participate in event design, they have control over the impacts and community benefits.

The reorientation from economic activities toward the service industry, primarily to tourism, represents one of the possible solutions for overcoming the problems with job loss in numerous industrial regions. Since many of these cities, some of which were industrial ones, did not have any significant attractions, it was concluded that the solution could be found in sustainable development of specific forms of cultural tourism, based on festivals and art tourism, as well as on attractions related to the industrial heritage of such cities [44].

As already mentioned, tourism was recognized by several countries over the last 30 years, especially developing ones, as one of the strong potential drivers of economic and sustainable development. Numerous econometric studies dealing with the dynamic effects of tourism development confirmed that tourism might contribute to overall economic development, especially in comparison with some traditional industries, particularly in economically unstable countries [45–50]. Therefore, it could be said that the multiplicative effects of tourism development on the infrastructure sector, together with the development of human resources, the development of the construction industry (in the form of building new hotel capacities), and the development of small and medium enterprises in the function of tourism development, represent a chance for restructuring industrial regions, such as Lazarevac and its surroundings.

When it comes to the mine surfaces that are no longer exploited and which represent the so-called *geo-touristic attractiveness*, there is a possibility of creating sustainable options for the devastated environment and a complex tourism product of industrial and geo-tourism [41]. There are numerous examples of production plants of global companies that attract several hundred thousand visitors per year. For instance, in Germany, great attention is oriented toward the industrial cultural heritage of the Ruhr region, with a 400-km-long cultural route related to industrial heritage, which includes an industrial heritage train and six museums of technical and social history of the Ruhr region [51–55].

The possibility for even faster development of industrial tourism is also reflected in the fact that postmodern tourists visit a destination with a specific goal of revealing something relevant to them, and not just because they have a lot of free time or because of a possible perception of a destination as a place of good entertainment. This reorientation of tourists' interest toward cultural attractions, including industrial attractions, contributed to the intentions of numerous development planners of tourism destinations to conduct measures for providing the development of industrial tourism [32,56,57].

The research was conducted on the territory of Lazarevac, a well-known mining region, which faces the process of restructuring in the upcoming period with a possible reduction in the number of employees of Kolubara mine. As a result, the sector of tourism entrepreneurship could be viewed as an opportunity for creating new jobs. Great potential for the development of various selective forms of tourism, ranging from cultural and industrial tourism to sport and rural tourism, highlights tourism as one of the most important sustainable development opportunities for this region. The

practices of economically developed countries show that the governments of these countries use a variety of incentive measures for entrepreneurship ventures [58]. These measures are also important for transitional societies, such as Serbia, where the national plan for regional development of this country is focused on direct stimulation of economic activity. According to this national plan, more than 60% of funds are focused on stimulating entrepreneurship development across the regions of Serbia [59]. This could particularly be regarded as a chance for starting entrepreneurship ventures in the sector of tourism. Different forms of local support, often related to social capital, as a product of various social relations between numerous social groups, could significantly facilitate the initiation of entrepreneurial activity. Therefore, the research related to entrepreneurship could also be viewed within the framework of the social network theory, primarily on the basis of the complexity of entrepreneurial venture and its connection with numerous actors in the local community [41,60–72].

A contemporary approach to the studies of entrepreneurship influenced the selection of appropriate questionnaires for the purpose of this research, including one which measures the level of residents' commitment to the overall local community [73–75]. Thus, this research relied on the so-called exogenous approach, oriented toward researching the intentions of the local community with regard to starting the entrepreneurial ventures, particularly practiced in recent years (in addition to an endogenous approach, primarily based on researching the individual characteristics of entrepreneurs).

## 3. Methods

### 3.1. Instrument

The survey research was conducted using a questionnaire, divided into several segments. The first segment dealt with the socio-demographic characteristics of the respondents; thus, the analysis of the sample structure was researched according to the following socio-demographic variables: gender, age, education, income, profession, and location of residence.

In the second segment of the questionnaire, the respondents were asked three sets of questions about the participation of the local population in the tourism industry, in order to evaluate the readiness of the local community members with respect to their involvement in tourism development, in different conditions regarding the implemented incentive measures within this sector. **Participation 1** covered the questions related to the respondents' current involvement in tourism development, or, more precisely, whether they were engaged in providing the services of accommodation, food, transportation, and similar at the time of conducting the survey. **Participation 2** covered the issues related to the intention of the respondents and their interests to be involved in the tourism sector in due time and under certain conditions. Finally, **Participation 3** comprised the respondents who expressed a certain level of readiness to be involved in the further development of tourism, in the case of already developed tourism in Lazarevac (as a result of strategic planning), compared to the current situation. The importance of the development of industrial tourism, as a form of cultural tourism, in promoting mining and industrial heritage was particularly pointed out.

The next segment of the questionnaire consisted of a questionnaire on the "acceptability of tourism development options" [76]. The respondents expressed their attitudes, ranging from 1 (not acceptable) to 5 (extremely acceptable), regarding different tourism development options, grouped into three factors: entertainment, attractions, and infrastructure.

Furthermore, the research was also conducted through another standardized questionnaire related to "attitudes toward tourism development" [77]. In this case, the respondents also evaluated the items based on a five-point Likert scale, ranging from 1 (completely disagree) to 5 (completely agree). These items were related to the respondents' general level of support for the development of tourism, as an important direction for the development of the entire local community. For example, the respondents were asked if they thought that tourism could play a vital role in the development of the local community, if they supported the development of new tourism facilities, or if they believed that tourism could be one of the most important industries within the local community in the future.

*3.2. Hypotheses*

Based on the literature review, the following hypotheses were formed:

**Hypothesis 1 (H1).** *Participation of the local population in tourism activity depends on the initiative and support of the state/municipality to a great extent, which could improve conditions for tourism development by using different measures.*

**Hypothesis 2 (H2).** *Local residents who are interested in participating in entrepreneurship in tourism give higher scores for the economic benefits of tourism development, compared to respondents who are not interested in participating in tourism development.*

**Hypothesis 3 (H3).** *Local residents who are interested in participating in entrepreneurship in tourism give higher scores for contribution of tourism to the quality of life within the destination, compared to respondents who are not interested in participating in tourism development.*

**Hypothesis 4 (H4).** *Local residents who are interested in participating in entrepreneurship in tourism give higher scores for supporting tourism development, compared to respondents who are not interested in participating in tourism development.*

**Hypothesis 5 (H5).** *The sense of community within the local population, as well as their high level of support for tourism development, might shape their perception of positive impact of tourism development on the quality of life.*

After collecting the data, the analysis was conducted using SPSS (17.0). Descriptive analysis was used for defining the sample characteristics. Questions related to the acceptability of tourism development options and attitudes toward tourism development were grouped into a smaller number of factors using factor analysis; then, hypotheses of the research were tested using a *t*-test and hierarchical regression analysis.

*3.3. Sample*

The research was conducted in Lazarevac, an industrial area with one of the largest open-pit mining surfaces in Europe. The sample consisted of 273 respondents, including 130 employees of Kolubara mine. The responses were collected through a standard pen-and-paper procedure. The respondents were informed that the research was anonymous, and they were asked to answer the questions sincerely. The study involved subjects who showed interest in participating, and the sample was considered appropriate. As the main focus of this research was on the attitudes of the entire local community related to restructuring industrial areas, data were collected not only from the respondents who are employed in Kolubara mine, but also from other members of the local community, or, more precisely, from those who were not employed by this company, but who worked in other companies, including private ones. The research also involved those who were unemployed or retired, and those who were still in the education process.

The majority of the respondents, 66.70% of the total sample, were female, while the remaining 33.30% were male respondents, as represented in Table 1. Based on the age structure, similar percentages of the respondents were recorded for the groups of respondents aged between 30 and 39 years (29.3%) and those aged 50 or more years old (29.7%). According to the research results, the majority of the respondents had a college or faculty degree (49.6%). The minority of the respondents, 8% of the total sample, finished elementary school, and the majority of them were currently students at a secondary school of economics. Based on the place of residence, the largest number of the respondents lived in urban areas, or, more precisely, in the town of Lazarevac (78.3%), while 21.7% of the respondents were from the surrounding settlements and villages.

**Table 1.** Respondents' socio-demographic characteristics.

| | | |
|---|---|---|
| Gender | Male | 33.3% |
| | Female | 66.7% |
| Age | Up to 29 years | 15.9% |
| | 30–39 years | 29.3% |
| | 40–49 years | 25.0% |
| | 50 years and more | 29.7% |
| Education | Primary school | 8.0% |
| | High school | 42.4% |
| | College (two- or three-year studies) | 15.2% |
| | Faculty (four-year studies) or master's studies | 34.4% |
| Place of Residence | Urban area | 78.3% |
| | Rural area | 21.7% |

## 4. Results

### 4.1. Participation of the Local Population in Tourism

Based on the research results, only 6.5% of the respondents were involved in tourism development, but 18.2% of them expressed their intention to get involved in this sector. It should also be noted that 46.2% of the respondents were interested in promoting the cultural and industrial heritage of the local community.

The research also considered the share of positive and negative responses in relation to potential forms of participating in the tourism activity. Figure 1 indicates the trend of the respondents' participation in tourism development, according to their attitudes. Only 18 respondents were currently involved in some kind of tourism activity, while 50 of them were interested in it. The biggest increase was registered in a situation when the respondents were asked if they would be ready to participate in tourism development in the case of municipal initiatives for the improvement of general conditions for tourism development. As many as 127 respondents gave a positive answer to this question, which allowed us to confirm H1.

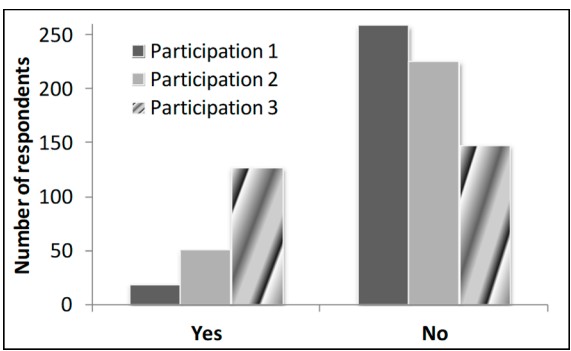

**Figure 1.** Trend of positive and negative responses related to participation in tourism development.

### 4.2. Factor Analysis—Respondents' Attitudes Regarding the Current State of Tourism Development Options and Attitudes toward Tourism Development

The analysis of the main components covered 16 issues related to acceptability of the options for tourism development. The value of the Kaiser–Meyer–Olkin's indicator exceeded the recommended level of 0.6 and, therefore, satisfactory results were achieved. Bartlett's test of sphericity reached statistical significance ($p = 0.000$), which justified the application of this analysis. After extraction of

the factors, the Varimax rotation method (with Kaiser normalization) was used. The identified factors were labeled as *entertainment*, *attractions*, and *infrastructure*. The detailed construction of each factor is represented in Table 2. It should be noted that the first factor, *entertainment*, consisted of the following items: trade development, visits to museums, construction of bars, and places for gambling. The second factor, *attractions*, contained the items related to development of parks and zoos, development of opportunities regarding sport and recreation, development of industrial tourism, visits to historical and cultural attractions, organization of various festivals, development of restaurants, and expansion of hotel capacities. The third factor, *infrastructure*, contained the following items: development of rural tourism, improvement of public transport, construction of campsites for tourists, expansion of private accommodation facilities, and development of various tourism services (including travel agencies and travel guides).

**Table 2.** Factor analysis: acceptability of tourism development options.

| Items | Entertainment | Attractions | Infrastructure |
|---|---|---|---|
| Construction of bars | 0.732 | | |
| Visits to museums | 0.731 | | |
| Trade development | 0.653 | | |
| Construction of places for gambling | 0.576 | | |
| Development of parks and zoos | | 0.664 | |
| Visits to historical and cultural attractions | | 0.593 | |
| Development of opportunities for dealing with sport and recreation | | 0.534 | |
| Development of industrial tourism | | 0.526 | |
| Organization of various festivals | | 0.523 | |
| Development of restaurants | | 0.493 | |
| Development of rural tourism | | | 0.734 |
| Improvement of public transport | | | 0.724 |
| Development of tourism services (travel agencies, travel guides) | | | 0.503 |
| Construction of campsites for tourists | | | 0.433 |
| Expansion of private accommodation facilities | | | 0.804 |

Furthermore, analysis of the main components covered 15 questions related to the attitudes of the local population toward tourism development. In this analysis, Kaiser–Meyer–Olkin's indicator exceeded the recommended value of 0.6. Also, Bartlett's test of sphericity reached statistical significance ($p = 0.000$), which justified the application of this analysis. After extraction of the factors, Varimax rotation with Kaiser normalization was used. The identified factors were labeled as *supporting tourism development* and *economic benefits of tourism development for the local community*. The detailed construction of each factor is represented in Table 3. Based on the research results, it should be noted that the first factor contained items related to the respondents' general level of support for the development of tourism as one of the important development directions of the local community. Moreover, the first factor also contained items related to the respondents' attitudes toward the following statements: tourism could play a vital role in the development of the local community; tourism employees in the local community should be more engaged in the promotion of tourism; the tourism industry should play a major economic role in the development of the local community. Another item which constituted the first factor suggested that respondents were proud when tourists came to see what their local community could offer, together with the last item indicating that respondents supported the development of new tourism facilities. The second factor contained items indicating the respondents' attitudes to whether they believed that rapid tourism development would increase the general development of the local community, and if tourism could be one of the most important industries in the local community in the future. Furthermore, the second factor contained items indicating that respondents showed respect toward tourists who visited them, that they believed that the overall

benefit of tourism development could overcome its possible negative impact, that the tourism industry could create new opportunities for employment, that tourism would contribute to a general increase in local revenue, and that the increase in the number of tourists might contribute to the development of the local community in general.

**Table 3.** Factor analysis: attitudes of the local community toward tourism development.

| Items | Supporting Tourism Development | Economic Benefits of Tourism Development for the Local Community |
|---|---|---|
| The local community should be involved in planning and managing tourism. | 0.814 | |
| Feeling of pride when tourists visit the local community. | 0.762 | |
| Tourism might have a vital role in the development of the local community. | 0.751 | |
| Tourism should have an important economic role in the development of the local community. | 0.719 | |
| Necessity of improvement in the domain of tourism promotion. | 0.716 | |
| Supporting tourism development as one of the main directions of development of the local community. | 0.694 | |
| Supporting the construction of new tourism facilities. | 0.686 | |
| Increase in local revenue. | | 0.853 |
| New employment opportunities in tourism. | | 0.837 |
| an increase in the number of tourists would provide benefits for the local community. | | 0.807 |
| The positive impacts of tourism are higher than the negative ones. | | 0.676 |
| Tourism development could influence the positive development of the local community. | | 0.619 |
| Tourism could be one of the most important industries in the future. | | 0.579 |
| Respecting the tourists. | | 0.573 |

The item indicating the respondents' opinions regarding their potential personal benefit from the development of tourism in the local community was excluded from the factor analysis. However, the respondents' answers to this question were dichotomized as follows: the respondents who answered with 1, 2, or 3 were considered to not see the personal benefit of tourism development, while those who answered with 4 or 5 thought they could personally benefit through tourism development. Accordingly, the results of the respondents' answers to the stated item were analyzed only in order to differentiate the respondents into two categories—those who considered that they could benefit through tourism development and those who did not.

In addition to factor analysis, research was also conducted by *t*-test analysis in order to identify differences in the arithmetic mean for the factor termed *economic benefits of tourism development for the local community* in the sub-sample of respondents who were interested in participating in tourism entrepreneurship (in conditions of developed tourism) and those who were not interested in participating (Table 4). The research results indicated that the respondents interested in participating in entrepreneurship in tourism provided a statistically higher score when evaluating the economic benefits of tourism development for the local community (4.35), compared to those respondents who were not interested in participating in tourism development (3.89), which resulted in the acceptance of H2. These results might suggest that a high level of awareness regarding the economic benefits of tourism development and the general contribution of tourism to the quality of life for the entire local

community were the main reasons shaping the respondents' interest in participating in entrepreneurial activities in the field of tourism. It is also important to point out that these respondents were aware of the importance of providing support for general tourism development.

**Table 4.** Economic benefits of tourism development for the local community according to the respondents' intentions to participate in entrepreneurship in tourism.

| | | Levene's Test for Equality of Variances | | *t*-Test for Equality of Means | | | | | | |
|---|---|---|---|---|---|---|---|---|---|---|
| | | | | | | | | | 95% Confidence Interval of the Difference | |
| | | F | Sig. | *t* | Df | Sig. (2-tailed) | Mean Difference | Std. Error Difference | Lower | Upper |
| Economic benefits of tourism development for the local community | Equal variance assumed | 8.683 | 0.003 | 4.669 | 272 | 0.000 | 0.45627 | 0.09773 | 0.26387 | 0.64867 |
| | Equal variance not assumed | | | 4.755 | 268.976 | 0.000 | 0.45627 | 0.09596 | 0.26734 | 0.64520 |

Note: F—F-value; Df—degrees of freedom; Sig.—statistical significance for *p*-value (0.001); Std.—standard.

We further applied the *t*-test analysis to determine the difference in arithmetic mean for the factor termed *contribution of tourism to the quality of life within the destination* in the sub-sample of respondents who were interested in participating in tourism entrepreneurship (in conditions of developed tourism) and those who were not interested (Table 5). The research results indicate that those respondents who were interested in participating in entrepreneurship in tourism (4.53) provided statistically higher scores when evaluating the *contribution of tourism to the quality of life in the destination*, compared to those respondents who were not interested in participating in tourism development (4.23), which resulted in the acceptance of H3.

**Table 5.** Contribution of tourism to the quality of life in the destination according to the respondents' intentions to participate in entrepreneurship in tourism.

| | | Levene's Test for Equality of Variances | | *t*-Test for Equality of Means | | | | | | |
|---|---|---|---|---|---|---|---|---|---|---|
| | | | | | | | | | 95% Confidence Interval of the Difference | |
| | | F | Sig. | *t* | Df | Sig. (2-tailed) | Mean Difference | Std. Error Difference | Lower | Upper |
| Contribution of tourism to the quality of life in the destination | Equal variance assumed | 11.124 | 0.001 | 3.422 | 273 | 0.001 | 0.30131 | 0.08805 | 0.12797 | 0.47466 |
| | Equal variance not assumed | | | 3.548 | 245.678 | | 0.30131 | 0.08493 | 0.13404 | 0.46859 |

Note: F—F-value; Df—degrees of freedom; Sig.—statistical significance for *p*-value (0.001); Std.—standard.

Furthermore, the research focused on the application of the *t*-test analysis to determine the differences in arithmetic means for the factor termed *supporting tourism development* in the sub-sample of respondents who were interested in participating in tourism entrepreneurship (in conditions of developed tourism) and those who were not interested (Table 6). According to the research results, those respondents who were interested in participating in entrepreneurship in tourism (4.44) provided statistically higher scores when evaluating the factor *supporting tourism development*, compared to those

respondents who were not interested in participating in tourism development (4.05), which resulted in the acceptance of H4.

**Table 6.** Supporting tourism development according to the respondents' intentions to participate in the entrepreneurship in tourism.

| | | Levene's Test for Equality of Variances | | *t*-Test for Equality of Means | | | | | | |
|---|---|---|---|---|---|---|---|---|---|---|
| | | | | | | | | | 95% Confidence Interval of the Difference | |
| | | F | Sig. | *t* | Df | Sig. (2-tailed) | Mean Difference | Std. Error Difference | Lower | Upper |
| Supporting tourism development | Equal variance assumed | 10.572 | 0.001 | 3.988 | 273 | 0.000 | 0.39284 | 0.09850 | 0.19893 | 0.58675 |
| | Equal variance not assumed | | | 4.080 | 267.147 | 0.000 | 0.39284 | 0.09629 | 0.20326 | 0.58242 |

Note: F—F-value; Df—degrees of freedom; Sig.—statistical significance for *p*-value (0.001); Std.—standard.

### 4.3. Hierarchical Regression Analysis

The obtained value of *R*-squared (determination coefficient) was 0.492, indicating that the model had a good fit, or, more precisely, that almost 50% of the variability of the dependent variable termed *contribution of tourism to the quality of life in the destination* could be explained by variables termed *supporting tourism development* and *sense of community among the local population*, as presented in Table 7.

**Table 7.** Coefficients in the hierarchical regression model.

| Model | Unstandardized Coefficients | | Standardized Coefficients | *t* | Sig. |
|---|---|---|---|---|---|
| | B | Std. Error | Beta | | |
| Constant | 1.375 | 0.192 | | 7.150 | 0.000 |
| Supporting tourism development | 0.518 | 0.041 | 0.584 | 12.567 | 0.000 |
| Sense of community among the local population | 0.215 | 0.044 | 0.229 | 4.919 | 0.000 |
| Dependent variable: contribution of tourism to the quality of life in the destination | | | | | |

Note: Sig.—statistical significance for *p*-value (0.001); Std.—standard.

Since the corresponding coefficients in the regression model were positive, increases in the positive evaluations of *sense of community among the local population* and *supporting tourism development* were followed by an increase in the evaluation of *contribution of tourism to the quality of life in the destination*. One of the items for evaluating the *sense of community among the local population* was as follows: "I am ready to participate in a job and cooperate with the people in my environment that will contribute to the general development of my surroundings". This refers to various voluntary actions related to arranging the appearance of the local community and educating the population regarding the importance of preserving the environment, which are among the most important factors in tourism development.

## 5. Discussion

The research results showed that the majority of the local population in Lazarevac were employed in the mining sector and associated work. Other sectors of economy were slightly represented and, therefore, restructuring should achieve significant diversification of economic activities in this industrial area. Reliance on the exploitation of coal for decades caused considerable inertness and a lack of

interest related to any initiative in the direction of reconstruction, not just among the local population, but also among the management structures of Lazarevac, despite the fact that numerous authors pointed out the importance of creating a stakeholder map. The importance of a stakeholder approach was emphasized in the study conducted by Perić, Đurkin, and Wise [26], which stated that, without the contribution of numerous stakeholders, further tourism development would be restricted, and that it could consequently result in a decrease in the positive social effects of tourism on the local community residents.

Although not too intensive, the intention of the respondents to participate in tourism activity (18.2%), compared to the current level of their involvement (only 6.5%), was a sufficient indicator that there is a certain basis for the optimistic attitudes of planners oriented toward the strategic development of Lazarevac. Based on the research results, education regarding the significance of tourism development and associated training would not only provide concrete assistance in realizing the projects, but could also encourage greater reorientation in the interest of the local community regarding participation in tourism development. The previous study showed that entrepreneurship education has a significant role in entrepreneurship activities, even in societies not primarily focused on entrepreneurship [78].

It should also be emphasized that the share of respondents who were interested in promoting the cultural and industrial heritage of the local community in an organized and planned manner was 46.2% of the total sample. The respondents clearly defined the conditions under which they would seriously consider their involvement in this activity. This is important to bear in mind, due to the fact that different authors indicated that local residents are important stakeholders. Accordingly, the support of members of the local community oriented toward tourism development could be a critical factor for determining the success of destinations [12].

The research results related to the respondents' gender structure showed that females expressed slightly higher interest to participate in the development of tourism in the local community (47.5% versus 43.5%). Based on the research results, education degree might be an encouraging factor for choosing to participate in tourism entrepreneurship. The fact is that respondents who studied tourism and had similar professions were predominant in the sample, followed by those with secondary education degrees in the context of different educational profiles. Both groups saw the opportunity for employment in the field of tourism more so than respondents from other categories of education.

The youngest respondents (up to 29 years of age) were the most interested in tourism. Their optimism in terms of realistic opportunities for developing the tourism sector on the local level was very pronounced. They also believed that this sector could generate significant revenue, together with providing a positive influence on the entire local community. Almost half of the respondents (47.7%) tended to see a future job career in the local community, where competent institutions already provided the necessary conditions for shaping the tourism sector in a good, stable, and profitable economy.

The profession of the respondents influenced their affinity toward tourism in two basic indicative findings. Firstly, the respondents who were in the permanent educational process (secondary-school and university students) were mostly open to new activities, and they considered tourism as a sector with enormous development potential. Secondly, respondents from different activities (production workers in Kolubara mine, employees in administration, and those already working in the service activities) expressed similar attitudes with regard to their readiness to participate in tourism activities. Urban (45%) and rural (50%) environments gave a positive basis for the development of tourism activity, without significant differences. Regarding the study on tourism development options, factorial analysis of the questionnaire components revealed that three factors could be clearly distinguished: entertainment, attractions, and infrastructure, which, in fact, represent acceptable options for tourism development at this location.

Furthermore, it could be said that younger females, who were professionally prepared for tourism activity, stood out as potential carriers of tourism development in Lazarevac. These individuals should also be open to permanent education and a constant process of learning and practical training. Their

economic characteristics indicated that their annual income was low, or that they were even completely devoid of a constant annual income; thus, tourism represents an attractive opportunity for their employment. All these features of tourism development can be used as a good and useful guideline when dealing with the strategic development of tourism activity. Also, all other results of statistical analyses provided solid grounds for further implementation of the project of shaping a new tourism product in Lazarevac.

This research included the attitudes of the local community regarding their participation in tourism. Two types of participation were established: unconditional and conditional (which involved participation in entrepreneurial activities in the field of tourism, in conditions of developed tourism infrastructure). A higher level of interest among respondents was recorded regarding conditional participation. Such a positive trend indicated a favorable social atmosphere for the development of tourism activity at an organized and planned level. Not only did the respondents demonstrate that they recognized the potential for the tourism development of Lazarevac and surrounding settlements, but they also clearly defined conditions under which they would seriously consider their involvement in this activity. Factor analyses of the questionnaires "acceptability of tourism development options" and "attitudes toward tourism development" were also conducted. Three factors were identified for the first questionnaire: entertainment, attractiveness, and infrastructure. The factor analysis of the second questionnaire identified the existence of two factors: *supporting tourism development* and the *economic benefits of tourism development for the local community*. By applying the *t*-test analysis, it was concluded that those respondents who were interested in participating in entrepreneurship in tourism provided statistically higher scores when evaluating the *economic benefits of tourism development for the local community*, *supporting tourism development*, and *contribution of tourism to the quality of life in the destination*, compared to those respondents who were not interested in participating in tourism development. These results might suggest that, among the reasons for shaping the interest of respondents to participate in entrepreneurial activities in the field of tourism, a high degree of awareness regarding the economic benefits of tourism development and the contribution of tourism to the quality of life for the entire local community was highlighted. However, it is also important to note that the respondents were mainly aware of the general importance of providing support for the development of tourism. Hierarchical regression analysis for independent variables *supporting tourism development* and *sense of community among the local population*, on the one hand, and dependent variable *contribution of tourism to the quality of life within destination*, on the other hand, showed good fitting of the model, according to the *R*-squared value, equal to 0.492. This indicates that almost 50% of the variability of *contribution of tourism to the quality of life in the destination* could be explained by independent variables *supporting tourism development* and *sense of community among the local population* (which also represents the social capital of the local community to some extent). All beta coefficients of independent variables were significantly positive, which means that increasing the sense of community and providing support for the development of tourism is are followed by an increase in the respondents' awareness regarding the contribution of tourism to the general quality of life in a specific destination. According to the study conducted by Nunkoo and So [15], if the members of a local community perceive that the impacts of tourism might improve their quality of life, they will be more open to increasing their efforts and their level of support for the development of the tourism industry.

## 6. Conclusions

There are several preconditions important for successful restructuring of Lazarevac and its surroundings in the direction of sustainable tourism development. It is necessary for the management structures in the destination to fully understand the general importance of the service sector as one of the global directions of economic development, with the development of tourism as a particularly important aspect. Contemporary global changes in the trends related to tourism demand could provide different opportunities for tourists to acquire new knowledge (for example, increased interest in cultural and industrial tourism) and to support tourism development in Lazarevac.

Investment policy should also be a function of tourism development, as it is often neglected, and it should also help domestic entrepreneurs in the form of providing favorable loans with longer repayment periods, which could encourage potential entrepreneurs to participate in entrepreneurial activities. Furthermore, education in the tourism sector must be an important element when planning tourism development, while improving the skills of the local population regarding their knowledge of foreign languages might create a positive environment for the reception of foreign tourists from different countries. By analyzing the tourism potential of Lazarevac and its surrounding area, it can undoubtedly be concluded that this is a destination that could be recognizable in a fastidious tourist market as a destination that offers a unique tourist product, attractive to both domestic and foreign tourists, on the basis of adequate financial support and public–private cooperation. The local community largely supports the development of tourism and expresses willingness to participate in entrepreneurial activities in the field of tourism, especially in conditions of a relatively developed tourism superstructure in Lazarevac and its surroundings. Tourism is one of those service sectors in which the empathetic relationships between service providers and users are significant predictors of the perceived quality of service, and it is, therefore, very important to explore the attitudes of the local population toward tourism development.

It was shown that, with the support of a wider community, and with a greater involvement of the local management structures, local residents could show interest to participate in entrepreneurial activities in the field of tourism, which could significantly contribute to the sustainable development and successful restructuring of the traditional industrial region. Practical significance of this research is reflected in the application possibilities of the gained results when planning sustainable tourism development, based on the stakeholder approach of harmonizing the interests of different segments of the local community.

Further studies could focus on researching cultural dimensions, as an important framework for entrepreneurial activities in the tourism sector. Also, it is important to research a wider institutional profile at the national level for starting entrepreneurial ventures related to sustainable tourism development.

**Author Contributions:** S.N. designed the research and applied the field survey; M.N.K. organized the databases and analyzed the data; M.N.K. and S.N. carried out statistical calculations for the results; M.D.P., M.M.R., and M.G. conceptualized the draft, supervised the writing of all paragraphs, and edited the final version of the manuscript; M.M. and M.Š. coordinated the work and technically prepared the manuscript for submission.

**Funding:** This research was funded by the Ministry of Education, Science, and Technological Development, Republic of Serbia, grant numbers 176020 OI and 47007 III.

**Acknowledgments:** This research was supported by the Ministry of Education, Science, and Technological Development, Republic of Serbia (Grant No. 176020 OI and 47007 III).

**Conflicts of Interest:** The authors declare no conflicts of interest.

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
