# Peer review of "Changes in Traditional Activities of Industrial Area toward Sustainable Tourism Development"

_sustainability, doi:10.3390/su11226189_

Round 1

Reviewer 1 Report

A very good paper of particular interest to many readers and policy makers

Very interesting subject of particular relevance to many regions across the world which are undergoing transformations in theri economies and local societies.

Well structured and scientifically sound

Author Response

We are expressing our gratitude to the reviewer for his/her positive attitude toward our study.

Reviewer 2 Report

This is an interesting paper, and it does offer a useful discussion around the literature and the impact of tourism. I think you could do a little more to bring in work on tourism and sense of community and impact of tourism on the community. There is some work you might consider published in this journal and recently to reference and draw some references from:

http://ejtr.vumk.eu/index.php/volume22/668-v22rp107

https://www.mdpi.com/2071-1050/8/12/1337/htm

These papers I feel will also help with your discussion.

I find the methods and the results to be clear, but the discussion is far too descriptive and needs some critical evaluation to be improved, I think you can take some insight from the two papers I suggest the relate to community to use/reference some ideas from these papers to enhance your discussion and the impact of your work so that other researchers can see the added value of your study.

Hope this helps, I suggest the above revisions to enhance the work

Author Response

We thank the reviewer for the literature suggestions; we included information from the two suggested studies in the introduction and discussion sections. Besides that, we added the research results of several more articles, dealing with the attitudes of the local community regarding tourism development (lines 59-63, 67-95, 106-113).

Reviewer 3 Report

I am glad to have a chance to review this research manuscript. This study aims to find effective socio-cultural factors of resident’s attitudes toward tourism development and to evaluate readiness according to the sentences (line 84-88). In my understanding, this study looks like a case study rather than a research article. My concerns are on the lack of theoretical discussion and weakness of research design to achieve the research goals. To improve this manuscript, I would like to provide several comments and suggestions.

1. [Research title]
I do not understand how this research title fits the research content. What is the definition of ‘industrial identity’? It seems that this study mainly focuses on resident’s perceptions regarding the expected town development ideas.

2. [Abstract]
The majority of the abstract is about current issues in this community. The authors need to clearly describe the purposes, research design, meaningful results, the significant value of this research.

3. [Introduction]
There is no research problem. The issues in this town are facts and plans for community development. However, why does this study need to investigate resident’s attitude? Have the authors found any problems regarding the research subjects? If so, this study needs to address the research problems to be solved in this research project.

4. [Introduction]
No literature review. In the method and results, this study uses several questions for factor analysis and regression analysis. If so, this study needs to not only deeply discuss definitions of major concepts/constructs for this research, but widely organize findings and argument in previous literature. I strongly believe that there are many existing research articles about the factors leading to resident’s attitude.

5. [Introduction]
After the line of 88 which is about research goals, the contents related to current issues in the community are redundant. I would recommend developing a literature review with these topics based on the theoretical discussion.

6. [Methods]
What is the rationale to classify three ‘participation’ groups? The reason is not clear.

7. [Methods]
Provide the references of the questionnaire questions. If not, explain how the authors developed the questions.

8. [Methods]
The manuscript includes five hypotheses. However, a hypothesis must be proposed from a literature review. The authors might use a ‘research question’ for the purpose, but the theoretical background is needed.

9. [Methods]
The first paragraph in [2.2 Procedure] should be moved to [2.3. Sample].

10. [Methods]
The current content [2.3. Sample] can be organized in a table.

11. [Results]
Line 261: “… to conclude that the H1 is confirmed”. I do not think these results can approve this conclusion. A hypothesis must be accepted after an appropriate test. Especially, Figure 1 shows that there are more responses to ‘No’ than ‘Yes.’ The approach or result is not good to test the H1.

12. [Results]
Factor analysis must provide basic outputs like reliability, communality, KMO test, and so on which are not presented in this manuscript. There are the same issues in regression analysis.

13. [Discussion]
This version does not fully discuss the results to answer the research purposes and hypotheses. The majority of the discussion is pretty descriptive based on demographic segments of respondents.

14. [Conclusion]
This conclusion should point out the value of this research findings and provide future research ideas to overcome limitations of this research.

I hope the comments are useful to improve your manuscript

Author Response

[Research title]
I do not understand how this research title fits the research content. What is the definition of ‘industrial identity’? It seems that this study mainly focuses on resident’s perceptions regarding the expected town development ideas.
Authors’ comment: In response to the comment of the reviewer, we proposed a new title for the Article, without including the term ‘industrial identity’ (lines 2-3).
[Abstract]
The majority of the abstract is about current issues in this community. The authors need to clearly describe the purposes, research design, meaningful results, the significant value of this research.
Authors’ comment: We deleted parts of the abstract that were related to current issues in the local community and we put the emphasis on the purpose of the research, as well as on the research design, meaningful results and significant value of the research (lines 26-30).
[Introduction]
There is no research problem. The issues in this town are facts and plans for community development. However, why does this study need to investigate resident’s attitude? Have the authors found any problems regarding the research subjects? If so, this study needs to address the research problems to be solved in this research project.

Authors’ comments: We added the parts related to the research problem within the Introduction section. We also indicated why there is a need for researching the residents’ attitudes (lines 67-95).

[Introduction]
No literature review. In the method and results, this study uses several questions for factor analysis and regression analysis. If so, this study needs to not only deeply discuss definitions of major concepts/constructs for this research, but widely organize findings and argument in previous literature. I strongly believe that there are many existing research articles about the factors leading to resident’s attitude.

Authors’ comment: We added the section on Literature review and included information on previous studies on the topic (lines 97-169).

[Introduction]
After the line of 88 which is about research goals, the contents related to current issues in the community are redundant. I would recommend developing a literature review with these topics based on the theoretical discussion.

Authors’ comments: These changes are related to the previous authors’ comment. We made reorganization within the whole Introduction section. Now, part of the text after the research goals is the Literature review section.

[Methods]
What is the rationale to classify three ‘participation’ groups? The reason is not clear.
Authors’ comment: We added explanation of three ‘participation’ groups (lines 178-180).
[Methods]
Provide the references of the questionnaire questions. If not, explain how the authors developed the questions.
Authors’ comment: We added the references for standardized questionnaires within this study Refs 76 and 77 (lines 190-195).

[Methods]
The manuscript includes five hypotheses. However, a hypothesis must be proposed from a literature review. The authors might use a ‘research question’ for the purpose, but the theoretical background is needed.
Authors’ comment: The new section on Literature review is now reorganized to form the background for the hypotheses of the research.
[Methods]
The first paragraph in [2.2 Procedure] should be moved to [2.3. Sample].
Authors’ comment: We moved the first paragraph from 2.2 Procedures to 2.3 Sample (lines 225-235) and we changed the title Procedure into the Hypotheses (line 202).

[Methods]
The current content [2.3. Sample] can be organized in a table.
Authors’ comment: We added the Table 1 related to socio-demographic characteristics of the respondents (lines 251-253).
[Results]
Line 261: “… to conclude that the H1 is confirmed”. I do not think these results can approve this conclusion. A hypothesis must be accepted after an appropriate test. Especially, Figure 1 shows that there are more responses to ‘No’ than ‘Yes.’ The approach or result is not good to test the H1.
Authors’ comments: Figure 1 shows that the most ‘’YES’’ answers and the least ‘’NO’’ answers were obtained for the Participation 3. Therefore, this confirms the H1 (lines 260-268).

[Results]
Factor analysis must provide basic outputs like reliability, communality, KMO test, and so on which are not presented in this manuscript. There are the same issues in regression analysis.

Authors’ comment: We added the information on these outputs (lines 275-278 and 292-295).

[Discussion]
This version does not fully discuss the results to answer the research purposes and hypotheses. The majority of the discussion is pretty descriptive based on demographic segments of respondents.        Authors’ comment: We added some changes within the Discussion section in order to address these reviewers’ comments (lines 389-394, 401-403, 407-410, and 480-483).

[Conclusion]
This conclusion should point out the value of this research findings and provide future research ideas to overcome limitations of this research.

Authors’ comment: We added information related to proposals for future research (lines 516-519).

Reviewer 4 Report

I very much appreciated reading this manuscript. Thank you.

The theme - sustainable tourism development - is very important and the approach which has been adopted by the authors  - locals' involvement and attitudes towards it - is also of great interest both to researchers and practitioners.

I find that debating sustainable options for tourism is imperative in today's world, and tourism, for sure, is one of the areas where discussion of future, courses of action and decision making is of high importance. In fact, today, when humankind is facing huge challenges concerning sustainable life on the planet, all people of the tourism system should be called to participate. Action is required, however public bodies, agencies and organizations need to develop a new governance framework. For this to be devised, implemented and managed, these entities need to understand territory and social-economic dynamics, and the psychology (so to say) of local communities. The time is for people - stakeholders - act jointly, instead of against each other.

Congrats.

The study is robust from the theoretical and methodological points of view, as I was satisfied for detecting the expected references (e.g. Andereck et al., 2000). The authors covered also literature on the specific topic - industrial heritage tourism, as a species of cultural tourism and coherently related this literature with the case presented of Lazarevac. Based on this, the authors frame intelligibly the research objectives and methodology.

I appreciated the sections on discussion and conclusions. Authors should emphasize the the importance of public investment on entrepreneurial businesses and education, as these are the most relevant managerial recommendations to tourism bodies and agencies. Perhaps these sections could also address and question governance models to be adopted at such areas that favour inclusion of the community in the planning of tourism.

My suggestions for improvement relate to:

(1) structure

Section 1 - Introduction should be, in my view, divided in two. One which would be the introduction to the research topic and the rationale of the paper; and only then - Section 2 on Literature Review. This would cover: (i) residents' attitudes towards (sustainable) tourism development and (2) industrial tourism. As it is, the introduction is too long and exceeds its function as it mixes different sections of an academic paper: one of introducing the paper and the other of presenting prior research.

(2) Content 

a. To replace the extended and detailed description of the questionnaire, particularly ll. 166-182, by attaching the questionnaire to supplementary materials and summarizing this description in the text of the manuscript. The reading becomes rather boring and difficult this way. Maybe there are alternative, more appealing, ways to write this part.

b. Indicate, please, the version of SPSS used for purpose of accuracy.

c. In section 2.3 Sample, please insert a table with the sample's characteristics. It is useful for the reader to consult the table in a quick view, if necessary.

Author Response

(1) Structure

Section 1 - Introduction should be, in my view, divided in two. One which would be the introduction to the research topic and the rationale of the paper; and only then - Section 2 on Literature Review. This would cover: (i) residents' attitudes towards (sustainable) tourism development and (2) industrial tourism. As it is, the introduction is too long and exceeds its function as it mixes different sections of an academic paper: one of introducing the paper and the other of presenting prior research.

Authors’ comments: We added the parts related to the research problem within the Introduction section (we divided the entire Introduction section, according to reviewer’s comment). We also indicated why there is a need for researching the residents’ attitudes (lines 67-95). Besides that, paper now contains the Literature review regarding the main constructs of the research, such as residents' attitudes towards (sustainable) tourism development and industrial tourism (lines 97-169).

(2) Content 

To replace the extended and detailed description of the questionnaire, particularly ll. 166-182, by attaching the questionnaire to supplementary materials and summarizing this description in the text of the manuscript. The reading becomes rather boring and difficult this way. Maybe there are alternative, more appealing, ways to write this part.

Authors’ comment: We shortened the description of the questionnaires and we are attaching the questionnaire, as part of supplementary materials.

Indicate, please, the version of SPSS used for purpose of accuracy.

Authors’ comment: We added the information related to the version of the SPSS used for the purpose of this research (line 219).

In section 2.3 Sample, please insert a table with the sample's characteristics. It is useful for the reader to consult the table in a quick view, if necessary.

Authors’ comment: We added the Table 1 related to socio-demographic characteristics of the respondents (lines 251-253).

Round 2

Reviewer 2 Report

I am happy with the revised version, I feel the authors improved the paper and made stronger connections to the literature as well at the start and in relation to findings